# An Adaptive and Automatic Power Supply Distribution System with Active Landmarks for Autonomous Mobile Robots

**DOI:** 10.3390/s24186152

**Published:** 2024-09-23

**Authors:** Zhen Li, Yuliang Gao, Miaomiao Zhu, Haonan Tang, Lifeng Zhang

**Affiliations:** 1School of Electrical Engineering and Automation, Nantong University, Nantong 226021, China; li.zhen799@mail.kyutech.jp; 2School of Engineering, Kyushu Institute of Technology, Kitakyushu 804-0015, Japan; gao.yuliang101@mail.kyutech.jp (Y.G.); tang.haonan920@mail.kyutech.jp (H.T.); 3School of Computer Science and Engineering, Changshu Institute of Technology, Suzhou 215500, China; zhumm@cslg.edu.cn

**Keywords:** mobile robot, active landmarks, omnidirectional contact electrodes, power supply distribution

## Abstract

With the development of automation and intelligent technologies, the demand for autonomous mobile robots in the industry has surged to alleviate labor-intensive tasks and mitigate labor shortages. However, conventional industrial mobile robots’ route-tracking algorithms typically rely on passive markers, leading to issues such as inflexibility in changing routes and high deployment costs. To address these challenges, this study proposes a novel approach utilizing active landmarks—battery-powered luminous landmarks that enable robots to recognize and adapt to flexible navigation requirements. However, the reliance on batteries necessitates frequent recharging, prompting the development of an automatic power supply system. This system integrates omnidirectional contact electrodes on mobile robots, allowing to recharge active landmarks without precise positional alignment. Despite these advancements, challenges such as the large size of electrodes and non-adaptive battery charging across landmarks persist, affecting system efficiency. To mitigate these issues, this research focuses on miniaturizing active landmarks and optimizing power distribution among landmarks. The experimental results of this study demonstrated the effectiveness of our automatic power supply method and the high accuracy of landmark detection. Our power distribution calculation method can adaptively manage energy distribution, improving the system’s persistence by nearly three times. This study aims to enhance the practicality and efficiency of mobile robot remote control systems utilizing active landmarks by simplifying installation processes and extending operational durations with adaptive and automatic power supply distribution.

## 1. Introduction

With the development of automation and smart industries, the demand for autonomous mobile robots has expanded due to the need to replace labor-intensive tasks and address labor shortages [1,2]. In industrial production processes, the automatic tracking function of mobile robots is a crucial technology [3]. Currently, due to the development of computing devices and artificial intelligence algorithms [4,5,6], current autonomous robots can now recognize surrounding objects, determine their own position, and track their route. In the field of industrial applications, the automatic tracking function of mobile robots primarily relies on artificial intelligence or passive markers [7,8], such as lines or barcodes set on the surrounding environment.

However, the high-performance intelligent algorithms and multiple detecting sensors on autonomous mobile robots require high-speed computational capability and high cost, which brings great burden for industrial production [9]. Anyway, autonomous mobile robots using passive landmarks often face a series of issues, such as the inability to flexibly change tracking routes [10], difficulties in system equipment deployment, and insufficient recognition accuracy. Therefore, in this study, we proposed a control method for autonomous mobile robots using active landmarks to implement the automatic tracking function for industrial applications. Based on the light signals emitted by active landmarks, mobile robots can accurately identify the markers and achieve precise route tracking. This system does not require the mobile robots to have high artificial intelligence computational ability. Additionally, the deployment of active markers [11] is very flexible, allowing for easy changes to the robots’ tracking routes [12].

In a factory production line environment, the surroundings are often unknown to mobile robots, making it a critical challenge for them to perceive their environment and navigate autonomously. Since GPS cannot be used for localization in indoor environments, robots must rely on sensors, artificial landmarks, and intelligent algorithms for navigation. After reviewing a substantial amount of recent research, we summarize that robotic autonomous navigation generally falls into three main methods:Artificial landmark-based navigation: Robots using artificial landmarks only require low-cost visual sensors for landmark recognition and do not rely on complex navigation algorithms. Ref. [13] rapidly explores and maps the environment in a distributed manner using existing landmarks and achieves exploration, localization, and navigation within the environment. Ref. [14] proposes a navigation framework that uses global planning to identify high-value landmarks, and then employs DRL for local navigation, resulting in improved efficiency, safety, and success in dynamic environments. In industrial production settings, artificial landmark navigation can effectively reduce the development cost of robots. However, the downside of this method is that once the landmark routes are deployed, they cannot be easily changed, limiting the flexibility of the robot’s navigation route adjustments.SLAM (simultaneous localization and mapping) using visual sensors: This technique uses monocular or stereo cameras and requires high computational resources, alongside demanding SLAM algorithms. Ref. [15] introduces a method combining 2D motion inference and deep features for robust object tracking in semantic SLAM, improving accuracy despite detection challenges and moving cameras. Ref. [16] presents a SLAM-based navigation system for Pepper robots in dynamic indoor environments, achieving efficient localization with an average error of 0.51 m. Compared to artificial landmark methods, SLAM-based robotic navigation does not yet achieve the same level of accuracy. Additionally, such robots involve high technical development costs, potentially imposing a financial burden on factory production.LIDAR-based navigation: Robots utilizing LIDAR sensors can scan and create precise indoor maps, achieving higher accuracy than visual SLAM algorithms. Ref. [17] proposes LSMCL using 3D LiDAR for accurate localization in dynamic environments, with experiments confirming its effectiveness and real-world applicability. Ref. [18] proposes artificial landmark assistance estimator (ALAE)-Gmapping, an optimized SLAM method using artificial landmarks and genetic algorithms, achieving improved mapping efficiency and a 9.25% increase in localization accuracy. However, LIDAR sensors are expensive and lack semantic information, complicating subsequent algorithmic recognition.

Given that robots often have limited perception and computational resources, and that industrial production processes must consider cost efficiency, this paper proposes a rechargeable and flexibly deployable mobile landmark system. This solution addresses the high cost and inflexibility issues of industrial robot navigation, providing a high-precision, low-cost, and easy-to-install active landmark solution for industrial production.

This study represents multiple battery-powered luminous landmarks in arbitrary locations of industrial production scenes, which the robot can recognize and follow, allowing for flexible route changes. These landmarks are rechargeable, enhancing the system’s versatility. The robot identifies and follows landmarks that blink regularly to achieve route tracking in industrial production processes. Since the active landmarks are battery-powered, their placement is flexible, simplifying system setup and facilitating easy modifications to travel routes. However, their battery-powered nature presents a challenge for continuous long-term operation.

To address the battery depletion issue of active landmarks and enable continuous operation, an automatic power supply system was proposed. This system equips the robot with a power supply device that charges the active landmarks while it moves. Typically, precise position control of the robot is required for power supply. However, in this system, the active landmarks are equipped with omnidirectional contact electrodes, which can be expanded to any size. These electrodes allow contact charging without precise position control, enabling the robot to charge the battery regardless of its orientation or position as long as it is placed over the electrode. The omnidirectional contact electrodes surround the active landmark and receive power from the robot’s power supply device. As the robot moves, it lowers the electrodes of the power supply device onto the omnidirectional contact electrodes of the active landmark, providing contact charging to prevent battery depletion.

In terms of equipment deployment costs and the sustainability of the system’s operation, we miniaturized the active landmarks and proposed a continuously processing power supply distribution system. The large size of the electrodes on the active landmark side results in high costs and significant installation effort. Additionally, the active landmarks themselves can become obstacles for the robot. Furthermore, there is a limit to the amount of battery that can be installed on the robot, and the system uniformly charges all landmarks, leading to the robot running out of power before completing the charging process. Consequently, battery levels among the active landmarks become imbalanced, leading to inefficiencies in system operation. To address these issues, it is necessary to simplify the active landmark system and adjust the power distribution from the robot to the active landmarks. Therefore, this study proposes altering the structure of the automatic power supply system to simplify the system on the active landmark side.

The contributions of this paper can be summarized as follows:We proposed the use of rechargeable active landmarks to enhance the route-changing flexibility of autonomous mobile robots. The omnidirectional contact electrodes were designed to improve system robustness and increase the efficiency of the automatic power supply. Additionally, we miniaturized the landmarks to reduce equipment costs in industrial processes.We developed an adaptive power supply distribution method to extend the system’s operational duration by efficiently distributing the robot’s limited power among multiple landmarks. The Ambient cloud service was employed as the central management system for monitoring each landmark’s power data, facilitating overall control and adaptive power distribution.Extensive controlled experiments were conducted on the active landmarks and the automatic power supply distribution system. The results demonstrated the effectiveness of our automatic power supply method and the high accuracy of landmark detection. Our power distribution calculation method can adaptively manage energy distribution, significantly enhancing the system’s persistence by nearly three times.

This paper is structured as follows: Section 2 demonstrates the innovation points and critical techniques of our proposed automatic power supply module and the adaptive power distribution calculation module. Section 3 conducts plentiful controlled experiments to prove the effectiveness of our proposed adaptive and automatic power supply distribution system. Section 4 makes a comprehensive analysis of the experiment results. Section 5 provides a brief conclusion of our work.

## 2. Materials and Methods

The proposed method addressed key issues of automatic navigation function for autonomous mobile robots using active landmarks in industrial applications. In Section 2.1, we demonstrated the integrated design of the adaptive and automatic power supply distribution system. In Section 2.2, we described the design of the automatic power supply module, including the circuit configuration of the active landmark and the robot sides. In Section 2.3, a power distribution calculation module that improves the persistence of mobile robot navigation was proposed.

### 2.1. Adaptive and Automatic Power Supply Distribution System

Conventional autonomous mobile robots recognize passive landmarks such as lines, barcodes, or surrounding obstacles, which do not emit or receive signals themselves. Due to the limitations of neural network algorithms on target detection accuracy, autonomous mobile robots can never reach 100% object recognition accuracy [19,20]. Moreover, issues include the time-consuming equipment installation and difficulty in altering routes. To address these challenges, we proposed a new control method using active landmarks that emit signals actively. These active landmarks periodically blink, and robots follow them for navigation, as illustrated in Figure 1. When Landmark 1 blinks, the robot recognizes and follows it, while the others remain off. Once the robot reaches Landmark 1, its light goes off, and Landmark 2 blinks. The robot then recognizes and follows this new light, repeating this process for navigation control. This system can definitely achieve a 100% recognition rate due to the active signaling nature of the landmarks.

Furthermore, changing the sequence of blinking landmarks allows for flexible route adjustments. Since landmarks operate on battery power, they can be freely relocated, facilitating easy changes in the robot’s path. Thus, the approach of previous studies offers the advantage of 100% recognition rate and flexible route modifications. However, ensuring continuous operation by preventing battery depletion is crucial for these systems, as a single landmark’s battery failure could disrupt the robot’s movement. Therefore, the systems emphasize that continuous operation is crucial to avoid battery depletion.

The overall processing framework of our proposed adaptive and automatic power supply distribution is illustrated in Figure 2, which consists of an automatic power supply module and a power distribution calculation module. The processing procedure for our proposed system is as follows: Initially, all landmarks send battery information to Ambient cloud computing platform at regular intervals, typically once every few minutes. Then, using the collected battery information of active landmarks, calculations for power distribution are performed on the computing platform. Next, the computed results are then transmitted back to Ambient cloud computing platform, where the mobile robots access and obtain the remaining work time of the next landmark. Finally, based on this information, each robot proceeds to supply specific power from the charging unit to the respective landmarks.

### 2.2. Automatic Power Supply Module

Considering the requirement for flexible route changes in autonomous robots in practical industrial applications, we implemented a rechargeable landmark solution. Therefore, an automatic power supply system needs to be developed to prevent the batteries of landmarks from running out. This system enables mobile robots to provide energy to the landmarks out of power during the operation of the system. Figure 3 illustrates the system’s configuration. As depicted in Figure 3, the automatic power supply system can be divided into the landmarks and the robot sides. On the mobile robot side, multiple 3 × 3 panels are combined and arranged to surround the landmark and are connected to it. These 3 × 3 panels are thin sheet-like receiving devices (omnidirectional contact electrodes) capable of contact charging without precise position control. In this system, they serve as the power supply device to charge the active landmark side.

This study installed the omnidirectional contact electrodes on the robot side to simplify and miniaturize the landmark design. Since only one or two power-supplying robots are needed to charge the landmarks, and considering that a large number of landmarks are required to build a large-scale autonomous robot navigation system in industrial production lines, it is more cost effective and less complex to place the omnidirectional contact electrodes on the robot side rather than on the landmarks.

Figure 4 is a circuit diagram of the automatic charging system, the left side is the circuit diagram on the power supply side (landmark side), and the right side is the circuit diagram on the charging side (robot side). On the power supply side, a switch using a transistor is attached, and power can be supplied as needed. The robot side constantly measures the battery charge voltage, and when the battery voltage drops below a threshold voltage, it sends a charge start command to the landmark side. Furthermore, when the voltage is charged above a threshold voltage, a charge stop command is sent to the robot side. During that time, the DC/DC converter supplies a constant output to the microcontroller unit (MCU) and the landmarks.

#### 2.2.1. Design of the Active Landmarks

The proposed system’s active landmark is illustrated in Figure 5. The active landmark has a cylindrical structure, as depicted in Figure 5, with a diameter of approximately 15 cm and a height of about 5 cm. On the top surface, two circular electrodes of Phosphor Bronze are set for receiving power, while the sides feature multiple LEDs that flash to guide the robot. This design allows the active landmark to emit signals through blinking LEDs, facilitating the robot’s navigation and control without requiring direct contact or precise alignment.

The internal circuit configuration of the proposed active landmark circuit layout is shown in Figure 6 and Figure 7. The active landmark is structured with a power supply circuit, battery with its control circuitry, MCU, LED, and communication modules.

The MCU controls the LED modules to guide the robot and communicate the battery status (current voltage) to the cloud service Ambient for IoT [21] data storage. The battery, equipped to support rapid charging, utilizes an electric double-layer capacitor. The power from the battery is converted using a DC/DC converter to drive the MCU, communication, and LED modules. The flowchart of the work processing is shown in Figure 8.

The electric double-layer capacitor serving as the battery is externally powered through two circular electrodes mounted on the top surface of the cylinder. The active landmark typically operates in deep sleep mode to conserve battery consumption. At specified intervals $t_{1}$, it wakes up once to transmit battery information (current voltage *V*) to Ambient. Furthermore, at another specified interval $t_{2}$, along with transmitting battery information, it initiates the blinking of LEDs to guide the robot. The timing of waking up from deep sleep for communication and initiating LED blinking for robot guidance can be customized for each landmark.

#### 2.2.2. Design of the Mobile Robot Side

The circuit configuration on the robot side in the proposed system is illustrated in Figure 9. The power supply unit integrated into the robot includes a battery, constant current and switch circuits, MCU, communication module, and omnidirectional contact electrodes. The battery is selected with a capacity suitable for autonomous robot operations, considering the robot’s movement requirements. The constant current conversion circuit is essential for rapid charging from the robot to the landmark.

The proposed omnidirectional contact electrode equipment is designed for contact charging, serving as a device for power supply and reception. It consists of nine square panels arranged in a 3 × 3 grid and internal circuitry, allowing power to be supplied by ensuring contact with any two of the nine electrodes. Figure 10a shows the round-shaped electrodes of the power-receiving target, and Figure 10b shows the electrode pattern on the power-supplying side. As shown in Figure 10a, the power-receiving target has a positive terminal and a negative terminal, both of which are round terminals. If the diameter of the round terminal is D and the minimum distance between the terminals is L, then the maximum distance between the terminals is L + 2D. As illustrated in Figure 10b, the power-receiving electrodes consist of square panels arranged at regular intervals.

If the length of one side of the panel is *l* and the minimum distance between adjacent panels is *d*, the diagonal distance of the square panel is $\sqrt{2}l$, the maximum distance between adjacent panels is $\sqrt{2}d$, and the distance between panels that are three apart is 2l+3d. To ensure that the positive and negative terminals of the round electrodes make contact with any two electrodes of the arbitrary direction contact electrode in any orientation, the electrode pattern is designed based on the conditions given by Equations (Equation 1)–(Equation 3).
(1)$$\begin{array}{c} \begin{array}{c} D>\sqrt[{}]{2d} \end{array} \end{array}$$
(2)$$\begin{array}{c} \begin{array}{c} L>\sqrt[{}]{2l} \end{array} \end{array}$$
(3)$$\begin{array}{c} \begin{array}{c} L+2D<2l+3d \end{array} \end{array}$$

The system circuit configuration is shown in Figure 11. As illustrated in Figure 11, the arbitrary direction contact electrode consists of a power supply circuit, an electrode selection circuit, an electrode contact detection circuit, and nine electrode panels. Each circuit is controlled by a microcontroller unit (MCU). Here is an explanation of the operation: First, the electrode selection circuit selects two electrodes to check for contact through the blue line. Next, the contact detection circuit detects whether contact has been made. If no contact is detected, the electrode panels are switched. Once contact is detected, power is supplied from the power supply circuit through the red line. Additionally, this system can be used as a power-receiving circuit for detecting contact, excluding the power supply circuit, and is used as a power-receiving circuit in the system.

The omnidirectional contact electrode device developed in this study is depicted in Figure 12. We integrated omnidirectional contact electrodes on the robot side as a mobile charging apparatus. Under normal circumstances, automatic charging systems require multiple omnidirectional contact electrodes per active landmark, leading to high costs due to the necessity for various units per landmark and across numerous landmarks. In contrast, our system eliminates the need for omnidirectional contact electrodes on the active landmark side for power reception, consolidating them into a single unit on the robot side for charging. This simplification helps reduce costs significantly.

#### 2.2.3. Operation of the Automatic Power Supply System

Using the omnidirectional contact electrodes mentioned above as the charging unit on the mobile robot, it is possible to charge the landmark from the robot without requiring precise positioning control. Figure 13 illustrates a flowchart that represents the operation of the entire charging circuit. The communication module receives information regarding power distribution from the IoT data storage cloud service Ambient. Based on this information, the MCU operates the charging switch and initiates rapid charging of the landmark through the omnidirectional contact electrodes with a constant current. The charging process terminates at the appropriate time to ensure efficient power distribution.

As mentioned above, omnidirectional contact electrodes allow contact-based charging in any orientation or position. Figure 14a,b illustrates the electrode numbers on the robot side as ‘1’ to ‘9’; while on the landmark side, positive and negative electrodes are also designated, as represented by two yellow circles. It provides two specific examples of contact-based charging. Figure 14a depicts the view of omnidirectional contact electrodes and the landmark from a top view. As shown in Figure 14a, when the omnidirectional contact electrode makes contact with the landmark side, electrode ‘4’ is recognized as the positive electrode and electrode ‘6’ as the negative electrode, and power is supplied. If there is a displacement in position or orientation, as shown in Figure 14b, electrode ‘5’ becomes the positive electrode and electrode ’3’ becomes the negative terminal, and power is supplied.

### 2.3. Adaptive Power Distribution Calculation Module

During the operation of the automatic power supply system, mentioned above, the power consumption of each active landmark varies. If the mobile robot performs an equal charging operation for each landmark, the active landmarks that consume more power will deplete their energy earlier, causing the entire autonomous mobile robot operation system to collapse [22,23]. To ensure that the automatic mobile robot can work continuously during industrial production, we have proposed an adaptive power distribution calculation module. This module adaptively allocates power to each active landmark, thereby extending the durability of the entire automatic charging system.

To improve charging efficiency, it is necessary to adjust the amount of power supplied from the robot to the landmarks from two perspectives. The first perspective is the usage frequency of the landmarks. Landmarks with higher usage frequency have shorter expected operating times, while those with lower usage frequency have relatively longer expected operating times. Therefore, in the proposed method, the power supply distribution is adjusted to equalize the operating times of the landmarks by increasing the power supply to frequently used landmarks and reducing it to less frequently used landmarks. By adjusting the power supply distribution in this way, the variation in operating times among the landmarks can be eliminated. The second perspective is the remaining battery level of the robot. In the proposed method, the power supply distribution is adjusted based on the amount of battery power available for the robot to use for charging. If the robot’s remaining battery level is high relative to the total power supply required for all the landmarks, more power is supplied. Conversely, if the robot’s remaining battery level is low, the amount of power supplied is reduced. This adjustment ensures that even if the amount of power the robot can supply to the landmarks is limited, there will be no variation in operating times among the landmarks.

Therefore, the schematic diagram of the proposed power distribution calculation module is demonstrated in Figure 15. The power supply distribution is calculated so that the operating time of the landmarks after charging is equal, based on the operating time and the limited amount of battery power the robot can use for charging. The post-charging operating time of the landmarks is set uniformly to $t_{T}$, and models for supplying power from the robot’s power supply device to one active landmark, two landmarks, and multiple landmarks are considered. The method for calculating the power supply distribution is then discussed.

#### 2.3.1. Model for the Single Landmark

Consider the case where the robot supplies power to a single landmark. Let $Q_{\mathrm{charge}}\left[C\right]$ be the amount of charge the robot can supply, $Q_{0}\left[C\right]$ be the amount of charge in the landmark before charging, and Q[C] be the amount of charge in the landmark after charging. According to the law of conservation of charge, Equation (Equation 4) holds.
(4)$$\begin{array}{c} \begin{array}{c} Q_{charge}=Q-Q_{0} \end{array} \end{array}$$

Here, let the capacitance of the landmark be C[F], the voltage before charging be $V_{0}\left[V\right]$, and the voltage after charging be $V\left(t_{l}-t\right)\left[V\right]$, where $t_{l}$ is the operating time after charging. In other words, V(t) is the voltage when the remaining operating time after charging is $t_{l}-t$. With these definitions, it can be transformed into Equation (Equation 5).
(5)$$\begin{array}{c} \begin{array}{c} Q_{charge}=C\left(V\left(t_{l}-t\right)-V_{0}\right) \end{array} \end{array}$$

Here, C[F], V(t)[V], and $V_{0}\left[V\right]$ are known values. Let the operating time after charging be $t_{T}$. By substituting $t_{l}-t=t_{T}$, Equation (Equation 6) holds.
(6)$$\begin{array}{c} \begin{array}{c} Q_{charge}=C\left(V\left(t_{T}\right)-V_{0}\right) \end{array} \end{array}$$

By solving this for $t_{T}$, the expected operating time after charging can be determined. Furthermore, using $t_{T}$, the amount of charge supplied to the landmark can be expressed by Equation (Equation 7).
(7)$$\begin{array}{c} \begin{array}{c} Q_{charge}=CV\left(t_{T}\right) \end{array} \end{array}$$

#### 2.3.2. Model for Two Landmarks

Next, consider the power distribution model for two landmarks. Let the first landmark be Landmark 1 and the second landmark be Landmark 2. Let $Q_{\mathrm{charge}}\left[C\right]$ be the amount of charge the robot can supply, $Q_{01}\left[C\right]$ be the amount of charge in Landmark 1 before charging, and $Q_{02}\left[C\right]$ be the amount of charge in Landmark 2 before charging. The amounts of charge after charging of Landmark 1 and Landmark 2 are $Q_{1}\left[C\right]$ and $Q_{2}\left[C\right]$, respectively. Since the total battery charge of the robot is distributed between Landmarks 1 and 2, the law of conservation of charge gives Equation (Equation 8).
(8)$$\begin{array}{c} \begin{array}{c} Q_{charge}=\left(Q_{1}-Q_{01}\right)+\left(Q_{2}-Q_{02}\right) \end{array} \end{array}$$

Here, let the capacitance of Landmark 1 be $C_{1}\left[F\right]$, the voltage before charging be $V_{01}\left[V\right]$, and the voltage after charging be $V_{1}\left(t\right)\left[V\right]$. Similarly, let the capacitance of Landmark 2 be $C_{2}\left[F\right]$, the voltage before charging be $V_{02}\left[V\right]$, and the voltage after charging be $V_{2}\left(t\right)\left[V\right]$. Let the capacitance of the landmark be C[F], the voltage before power supply be $V_{0}\left[V\right]$, and the voltage after power supply be $V\left(t_{l}-t\right)\left[V\right]$.

Here, $t_{l}$ is the operating time after charging. Thus, $V_{1}\left(t\right)$ and $V_{2}\left(t\right)$ are the voltages when the remaining operating time after charging is $t_{l}-t$. With these definitions, this can be transformed into Equation (Equation 9).
(9)$$\begin{array}{c} \begin{array}{c} Q_{charge}=C_{1}\left(V_{1}\left(t\right)-V_{01}\right)+C_{2}\left(V_{2}\left(t\right)-V_{02}\right) \end{array} \end{array}$$

Furthermore, let $C_{1}\left[F\right]$, $C_{2}\left[F\right]$, $V_{1}\left(t\right)\left[V\right]$, $V_{2}\left(t\right)\left[V\right]$, $V_{01}\left[V\right]$, and $V_{02}\left[V\right]$ be known values. Furthermore, let the operating time after charging for both Landmark 1 and Landmark 2 be uniformly $t_{T}$. By substituting $t_{l}-t=t_{T}$ into the equations above, Equation (Equation 10) can be derived.
(10)$$\begin{array}{c} \begin{array}{c} Q_{charge}=C_{1}\left(V_{1}\left(t_{T}\right)-V_{01}\right)+C_{2}\left(V_{2}\left(t_{T}\right)-V_{02}\right)) \end{array} \end{array}$$

By solving this for $t_{T}$, the operating time after charging can be determined. Therefore, the amounts of charge supplied to Landmark 1 and Landmark 2 are given by Equation (Equation 11) and Equation (Equation 12), respectively.
(11)$$\begin{array}{c} \begin{array}{c} Q_{1}=C_{1}\left(V_{1}\left(t_{T}\right)\right) \end{array} \end{array}$$
(12)$$\begin{array}{c} \begin{array}{c} Q_{2}=C_{2}\left(V_{2}\left(t_{T}\right)\right) \end{array} \end{array}$$

#### 2.3.3. Model for Multiple Landmarks

Consider the model where there are *N* landmarks. Let the *n*-th landmark be Landmark *n* (for n=1,2,…,N). Let the amount of charge the robot can supply be $Q_{\mathrm{charge}}\left[C\right]$, and the amount of charge in Landmark *n* before charging be $Q_{0n}\left[C\right]$. Under these conditions, the amount of charge $Q_{n}\left[C\right]$ supplied to Landmark *n* can be determined. Since the total battery charge of the robot is distributed to all landmarks from 1 to *N*, the law of conservation of charge gives Equation (Equation 13).
(13)$$\begin{array}{c} \begin{array}{c} Q_{charge}=\sum_{n=1}^{n=N}\left(Q_{n}-Q_{0n}\right) \end{array} \end{array}$$

Here, let the capacitance of Landmark *n* be $C_{n}\left[F\right]$, the voltage before charging be $V_{0n}\left[V\right]$, and the voltage after charging be $V_{n}\left(t\right)\left[V\right]$. Note that $t_{l}$ is the operating time after charging. Thus, $V_{n}\left(t\right)$ represents the voltage when the remaining operating time after charging is $t_{l}-t$. With these definitions, it can be transformed into Equation (Equation 14).
(14)$$\begin{array}{c} \begin{array}{c} Q_{charge}=\sum_{n=1}^{n=N}C_{n}\left(V_{n}\left(t_{l}-t\right)-V_{0n}\right) \end{array} \end{array}$$

Furthermore, let $C_{n}\left[F\right]$, $V_{n}\left(t\right)\left[V\right]$, and $V_{0n}\left[V\right]$ be known values. By substituting $t_{l}-t=t_{T}$, where $t_{T}$ is the operating time after charging, Equation (Equation 15) can be derived.
(15)$$\begin{array}{c} \begin{array}{c} Q_{charge}=\sum_{n=1}^{n=N}C_{n}\left(V_{n}\left(t_{T}\right)-V_{0n}\right) \end{array} \end{array}$$

By solving this for $t_{T}$, the operating time after charging can be determined. The amount of charge supplied to all *N* landmarks is given by Equation (Equation 16).
(16)$$\begin{array}{c} \begin{array}{c} Q_{n}=C_{n}\left(V_{n}\left(t_{T}\right)\right) \end{array} \end{array}$$

#### 2.3.4. Acquisition and Management of Parameters

The method for adjusting the power distribution when there are multiple landmarks has been shown. However, the method for obtaining the known parameters $C_{n}$, $V_{n}\left(t_{l}-t\right)$, and $V_{0n}$ has not been mentioned. This section discusses these parameters, which are considered known. $C_{n}$ is the capacitance of the landmark’s battery and is measured in advance for each landmark. $V_{n}\left(t_{l}-t\right)$ is the relationship between operating time and voltage, which is also determined beforehand. The method for obtaining $V_{n}\left(t_{l}-t\right)$ is described as follows. First, measure the change in voltage over time $V_{n}\left(t\right)$ as shown in Figure 16a. $V_{n}\left(t\right)$ is measured by running the landmark in a trial operation that mimics the desired actual operation. By converting $V_{n}\left(t\right)$ using the operating time $t_{l}$, $V_{n}\left(t_{l}-t\right)$, as shown in Figure 16b, can be obtained.

$V_{0n}$ is the real-time battery voltage sent from the landmark when the system is actually operating. The landmark measures its battery voltage at intervals determined by the system administrator and sends these data to a personal computer (PC) through a cloud service.

In this study, the pre-charging voltages of numerous active landmarks are obtained and used in the PC-side calculations for charging allocation adjustment. To achieve this, it is convenient to centrally manage the data. For this centralized management, data are sent from the landmarks and managed using a cloud service called Ambient. Ambient is a cloud service for IoT data visualization provided by Ambient data corporation. It can accumulate and visualize sensor data sent from microcontrollers, allowing for the easy creation of IoT systems. In Ambient server, data sent from a microcontroller is managed in units called “channels”. Each user can create up to eight channels, and up to eight data points can be sent and graphed per channel. The sent data can be downloaded as comma-separated value (CSV) files. However, there are usage limitations. The total number of data points that can be sent to all channels combined in one day is limited to a maximum of 3000, and the data retention period is one year. In the proposed method, Ambient cloud service is used to achieve simple centralized management of data.

## 3. Results

Extensive controlled experiments have been conducted to demonstrate the effectiveness of our proposed methods. In Section 3.1, we first verify the practicality of omnidirectional contact electrodes in our designed active landmarks. Additionally, we show that the proposed automatic power supply system can also facilitate the navigation function of mobile robots. In Section 3.2, we provide a detailed implementation of the controlled experiments, confirming that our power supply distribution method enhances the system’s operational durability.

### 3.1. Experiments and Results of Automatic Power Supply Method

#### 3.1.1. Operational Verification for Omnidirectional Contact Electrodes

We provided validation of the operation effectiveness of the proposed omnidirectional contact electrodes mounted on the robot side in this experiment. As shown in Figure 17a, two of the nine omnidirectional contact electrodes (numbered 1 to 9) on the mobile robot side were brought into contact with the two circular electrodes on the active landmark side. If the connection is established, the LEDs on the landmark side light up, as depicted in Figure 17b. We exhausted all 72 possible electrode contact combinations by repeating the aforementioned experimental steps to verify the connection status in every pattern, and the experimental results were presented in Figure 18.

As depicted in Figure 18, the anode of the landmark is denoted as “+”, the cathode as “−”, and the electrode panels of the omnidirectional contact electrodes are numbered 1 to 9, consistent with Figure 17. For instance, if contact and successful power transfer are confirmed with electrode 1 on the “+” terminal and electrode 2 on the “−” terminal of the landmark, a circle is marked at the intersection of row 1 and column 2 in Figure 18. As evident from Figure 18, all patterns successfully established a connection. This confirms that the omnidirectional contact electrodes can be effectively applied in our proposed automatic power supply method.

#### 3.1.2. Detection of Active Landmarks

In this experiment, we verified the detection capability of active landmarks under a miniaturized design. Eight red LEDs installed on the side of the landmarks were set to blink at a cycle of 400 ms to test if the red light could be detected. Each LED had a luminous intensity of 330 mcd and a half angle of 30°. Figure 19 shows the process of recognizing the landmark using a camera installed on the robot. As illustrated in Figure 19, the camera was mounted at a height of 55 cm from the floor of the robot, and the detection capability was tested at distances ranging from 1 m to 15 m.

As examples of the results, the detection outcomes at vertical distances of 1 m and 15 m from the camera are shown in Figure 20a,b. We tested all distances at half-meter intervals, and it was demonstrated that the landmarks could be recognized at all distances from 1 m to 15 m with 100% accuracy. This detection range is sufficient for the practical use of the autonomous mobile robot control system in industrial applications.

#### 3.1.3. Error Analysis of Navigation

The charging contact range between robots and landmarks is crucial, as it impacts the control position accuracy requirements for robots. In this study, the landmarks were designed with a diameter of 15 cm and a height of 5 cm. The omnidirectional contact electrodes on the power supply side of the robots were installed at a height of approximately 8 cm, ensuring a good possibility of alignment with the landmarks. Thanks to distinctive identifiers, such as LED blinks for navigation, our robots can achieve nearly 100% accuracy in reaching the target landmarks. To address contact errors between the robots and the landmarks, we designed omnidirectional contact electrodes to enhance the contact success rate, which has proven effective in practical tests.

We define the contact range as the distance between the center points of the power supply device on the robot and the corresponding landmark. The optimal contact state occurs when the power supply device on the robot and the power-receiving device on the landmark are perfectly aligned. In other words, the center point of the power supply device on the robot and the center point of the landmark should coincide. Additionally, the center points of the round electrodes on the landmarks and the square electrodes at positions “4” and “6” should also overlap perfectly. In this ideal scenario, as depicted in Figure 21a, landmarks side represented by yellow circles and power supplying side perfect overlap, the distance between landmark and power supply devices is 0.

The maximum contact range must ensure that both electrodes on the landmark are within the coverage area of all nine electrodes on the power supply device. In the farthest range scenario, one of the electrodes on the landmark may be at the edge of the power supply device’s range in the horizontal direction. This is illustrated in Figure 21b. The sides of the square electrodes measure 4 cm, and the diameter of the circular electrodes is 3 cm. The maximum contact range is 3.5 cm.

Therefore, with a navigation error ranging from 0 cm to 3.5 cm, our autonomous robot can successfully charge at the landmarks. Thanks to the active optical sensors on the landmarks, our autonomous charging robot can localize the landmarks with nearly 100% accuracy. Thus, this navigation error range fully meets the requirements.

### 3.2. Verification of the Adaptive Power Distribution Calculation Method

#### 3.2.1. Experiment Implementation

Adjusting the power distribution requires information on the voltage relative to the operating time of each landmark. Therefore, in this experiment, we measured the time variation in the battery voltage during the operation of the landmarks in advance. As shown in Figure 22, “Landmark 1” is referred to as “L1”, and “Landmark 2” is referred to as “L2” in the following description.

L1 is assumed to be a rarely used landmark, while L2 is assumed to be a frequently used landmark. Consequently, the operating time of L2 is shorter than that of L1. The basic operations involve a deep sleep and restart period of 5 min for L1, and 1 min for L2. Each landmark is primarily in deep sleep mode to conserve battery power. L1 restarts once every 5 min to measure the battery voltage and send it to Ambient server, while L2 spends 1 min. Additionally, the landmarks blink at regular intervals to guide the robot. The guidance actions for the robot by each landmark are shown in Table 1.

L1 sends the battery voltage and blinks the robot guidance LEDs for 20 s at a 300 ms cycle every 30 min. L2 sends the battery voltage every minute and blinks the robot guidance LEDs for 20 s at a 400 ms cycle. The changes in voltage during operation over time for both L1 and L2 were measured every second and plotted, as shown in Figure 23a,b.

Figure 23a is a graph of the elapsed operating time and battery voltage for L1. The landmark was fully charged to 9 V and operated until it reached its lower operating voltage limit. The lower operating voltage limit is defined as the time when the battery information can still be transmitted. The lower operating voltage limit for L1 was 4.05 V, and the operating time was 3349 s.

Figure 23b is a graph of the elapsed operating time and battery voltage for L2. It was fully charged to 9 V and then discharged to its lower operating voltage limit. The lower operating voltage limit for L2 was 3.01 V. The lower operating voltage limit is lower because the deep sleep mode is longer compared to L1, which helps reduce power consumption. The operating time was 3063 s. Converting the voltage change over time during operation to the relationship between the remaining operating time and voltage, resulted in Figure 23c,d. Figure 23c,d represent the remaining operation time and voltage relationship of L1 and L2, which can be transformed from Figure 23a,b according to Equation (Equation 14).

#### 3.2.2. Verification of Power Distribution Adjustment

In this experiment, we verified whether it is possible to extend the operating time of the system by adjusting the power distribution when the amount of electric power available for the robot is limited. Specifically, we operated two landmarks with the same movements, as shown in Table 1 from a full charge. Each landmark transmits battery information at regular intervals. Figure 24 shows the battery information sent to Ambient server, converted to a percentage display.

The adjustment of power distribution is assumed to be carried out during the operation of the landmarks. Therefore, the power distribution adjustment is performed 2754 s after the start of operation. It is assumed that the power supply is provided to L1 and L2 in sequence using two power supply methods, with the power supply time being sufficiently short relative to the operating time. The first method attempts to fully charge the two landmarks without adjusting the power distribution. The second method involves adjusting the power distribution.

The battery information after 2754 s from the start of operation is shown in Table 2, derived from Figure 24a,b. Before charging, the battery level of L1 is 47.6%, and the battery level of L2 is 49.3%. The amount of battery power that can be supplied from the robot’s power supply device is 237 [C]. This amount corresponds to 15% of the 22 Ah battery capacity assumed to be mounted on the robot, distributed across 40 landmarks, and is enough for two landmarks. It is not sufficient to fully charge the two landmarks. Under these conditions, the results of the power distribution adjustment calculation by the program are shown in Table 2. The power is supplied until the battery levels reach the post-charge levels from the pre-charge levels. The subsequent operating time and battery levels were then examined.

#### 3.2.3. Comparison Results of Two Landmarks

The curves in Figure 25 illustrate the relationship between elapsed time and battery level for L1 and L2, respectively, and whether the system is powered with the power distribution calculation module or not.

We combined the four graphs of L1 and L2 with and without power distribution adjustment, as shown in Figure 25. The blue line represents L1 without power distribution adjustment, and the purple line represents L2 without power distribution adjustment. The red line represents L1 with power distribution adjustment, and the orange line represents L2 with power distribution adjustment.

From this graph, the system’s operational time can be determined. Operational time, in this case, refers to the duration during which both landmarks are operational. In the graph, the operational time can be identified as follows: In the case without power distribution (purple graph), the system’s operational time ends at 393 s when the battery level reaches 0 (purple graph). In the case of power distribution (orange graph), the system’s operational time ends at 1220 s when the battery level reaches 0 (orange graph). Comparing the operational times between adjusting power distribution and not adjusting it, the graph resembles Figure 26.

As seen from Figure 26, without adjusting the power distribution, the operational time for L1 when attempting to charge fully was 2777 s, and for L2, it was 393 s. The system’s operational time is defined as the duration during which both landmarks are operational, which, in this case, is 396 s. The shorter operational time for L2 when attempting full charge is because L1 was fully charged first, depleting the robot’s charge available for supplying L2 sufficiently. In contrast, when adjusting the power distribution, the operational time for L1 was 1234 s, and for L2, it was 1220 s, resulting in a system operational time of 1220 s. Although both landmarks were not fully charged, the system’s operational time was extended by 827 s compared to the attempt to fully charge them. Therefore, adjusting the power distribution extended the operational time with a limited power supply. It indicates that adjusting the power distribution improves the efficiency of the power supply.

## 4. Discussion

This study conducted several experiments to validate the effectiveness of our proposed adaptive and automatic power supply distribution system to enhance the operational efficiency and flexibility of autonomous mobile robots in industrial applications. Firstly, the effectiveness of omnidirectional contact electrodes was verified. The experiment tested 72 possible contact combinations, all of which successfully established a connection. This result confirms that the omnidirectional contact electrode system can reliably function in practical operations, as it eliminates the need for precise positional control, thereby simplifying the charging process. In the active landmark detection experiment, the ability to detect landmarks through the blinking signals of red LEDs was assessed. The results showed that landmarks could be successfully recognized at distances ranging from 1 m to 15 m with 100% accuracy. This indicates that the miniaturized landmarks can be reliably detected within practical distance ranges, demonstrating their applicability in the autonomous mobile robot system. To verify the potential of adaptive power distribution calculation methods to improve power supply efficiency and system persistence, comprehensive controlled experiments were conducted. Without power distribution adjustment, the system’s operational time was only 396 s because the robot prioritized fully charging Landmark 1, leading to insufficient charging for Landmark 2. In contrast, with power distribution adjustment, the system’s operational time significantly extended to 1220 s. Our power distribution calculation method can adaptively conduct energy distribution and improve the system persistence by almost 3 times. Although neither landmark was fully charged, the adjusted power distribution notably prolonged the system’s operational time, proving the effectiveness of power distribution adjustment under limited power conditions.

Overall, the experimental results of this study demonstrate that through the automatic power supply and optimized power distribution modules, the operational efficiency and flexibility of the autonomous mobile robot system can be significantly enhanced. Specifically, the power distribution adjustment method effectively extended the overall operational time under limited power conditions, validating its feasibility and advantages in practical industrial applications. These improvements not only reduce the equipment deployment costs but also simplify the installation process, enabling autonomous mobile robots to operate more efficiently across a broader range of industrial environments.

## 5. Conclusions

This study aimed to develop an automatic charging system for active landmarks that is cost effective, easy to install, and efficient in adaptive power delivery. Traditional methods of route planning for robots have limitations in flexibility and efficiency, leading to the need for innovative solutions. Our proposed approach utilizes miniaturized, battery-powered active landmarks, which can be flexibly placed and easily detected by robots, thus facilitating dynamic route adjustments. The system’s key innovation is the automatic power supply mechanism with omnidirectional contact electrodes and active landmarks. This design allows the robot to charge landmarks without precise positional alignment, thereby ensuring continuous operation. Despite initial challenges, such as the large size of landmarks and the need for system-enduring processing, our research focused on miniaturizing the landmarks and optimizing adaptive power distribution to address these issues. Extensive experiments collectively indicate that the proposed adaptive and automatic power supply distribution system successfully minimizes installation constraints and costs while improving charging efficiency and operational duration. This research highlights the practical application of rechargeable active landmarks and omnidirectional contact electrodes to provide a significant step forward in flexible and efficient route planning, and the power distribution calculation method to charge landmarks adaptively. Future work will aim to further refine the power supply system and explore additional methods to enhance the adaptability and robustness of autonomous mobile robot systems in various industrial environments. In conclusion, the integration of the proposed automatic power supply and adaptive power distribution calculation modules addressed critical issues of route flexibility and efficiency in autonomous mobile robots, offering a practical and innovative solution for industrial applications.

## Figures and Tables

**Figure 1 sensors-24-06152-f001:**
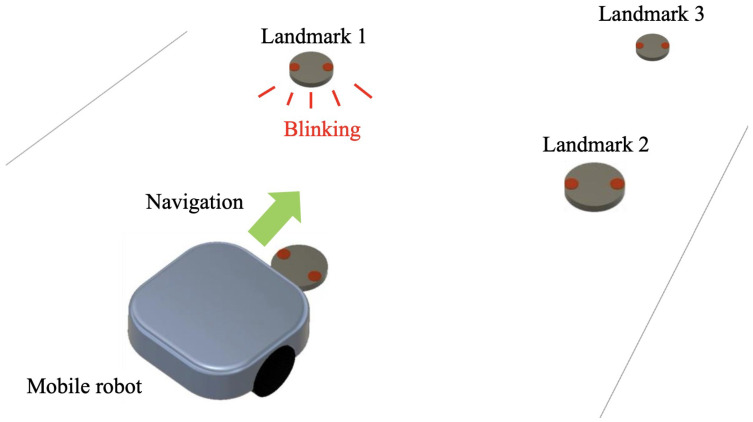
Operational illustration of autonomous mobile robots using active landmarks.

**Figure 2 sensors-24-06152-f002:**
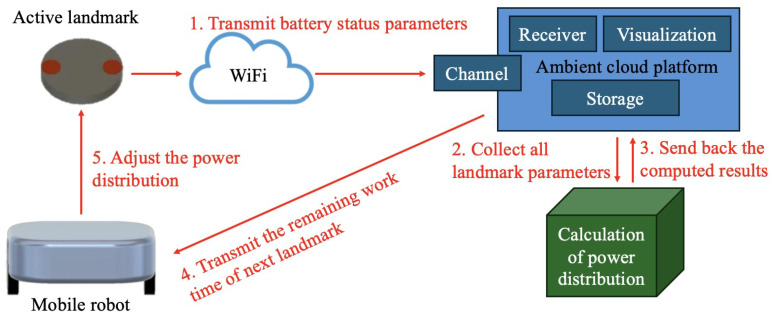
Overall framework of the adaptive and automatic power supply distribution system.

**Figure 3 sensors-24-06152-f003:**
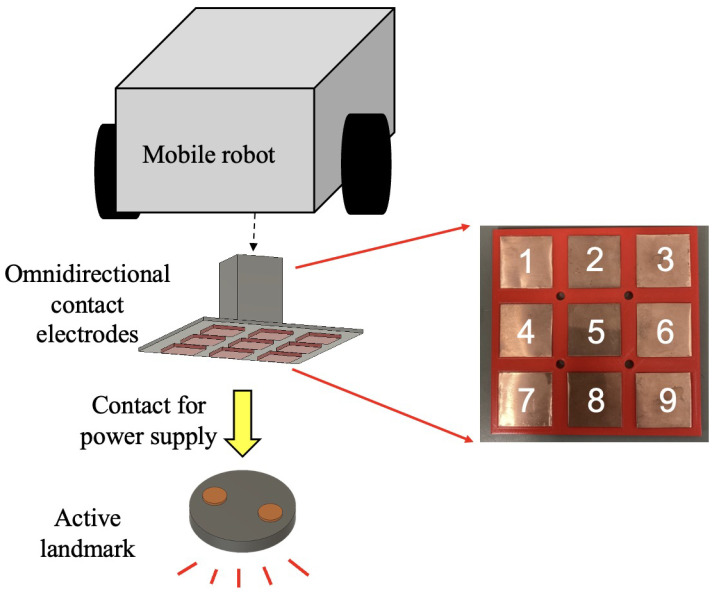
Active landmarks equipped with an automatic power supply system.

**Figure 4 sensors-24-06152-f004:**
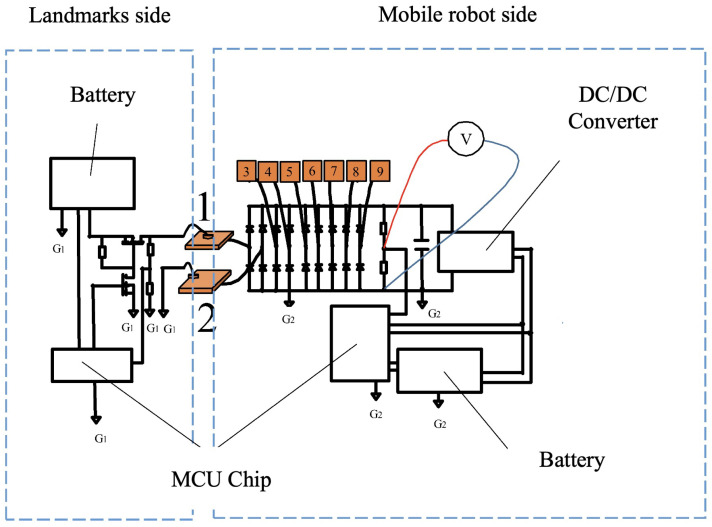
Processing scenes of the proposed automatic power supply system.

**Figure 5 sensors-24-06152-f005:**
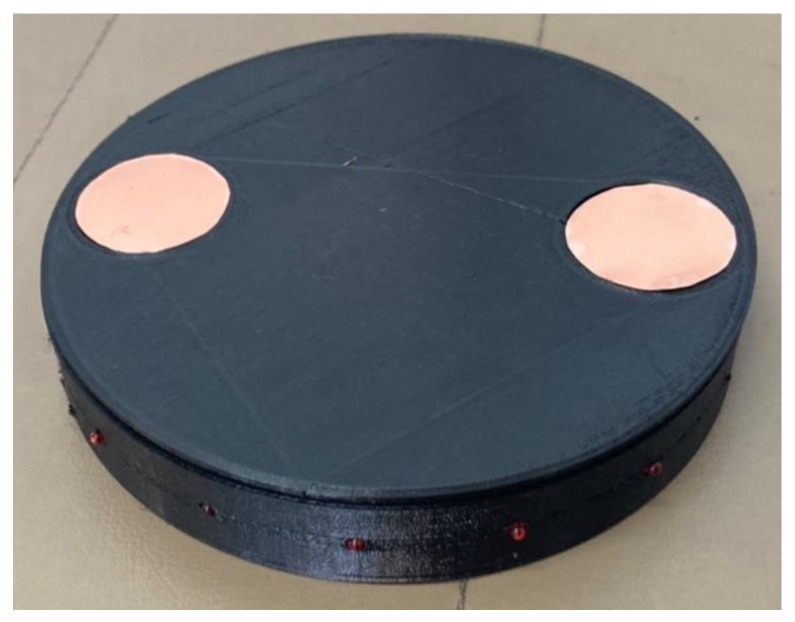
Active landmark equipment.

**Figure 6 sensors-24-06152-f006:**
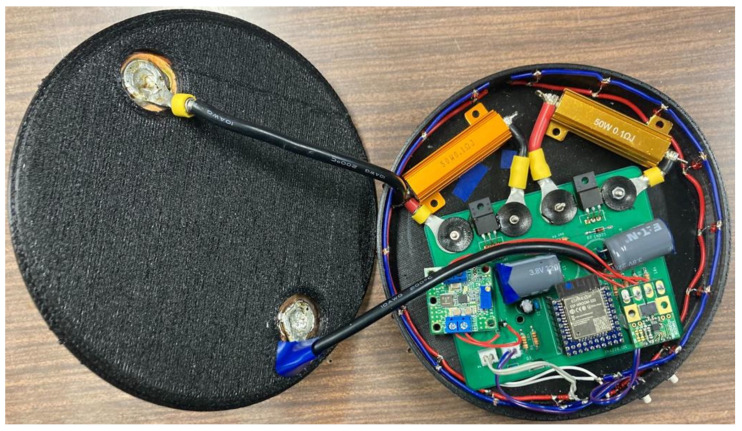
Interior circuit of active landmarks.

**Figure 7 sensors-24-06152-f007:**
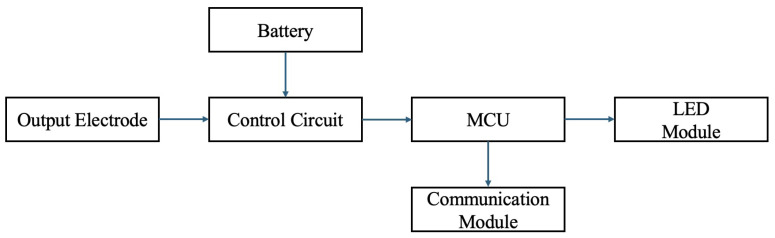
Constitution of the active landmark circuit.

**Figure 8 sensors-24-06152-f008:**
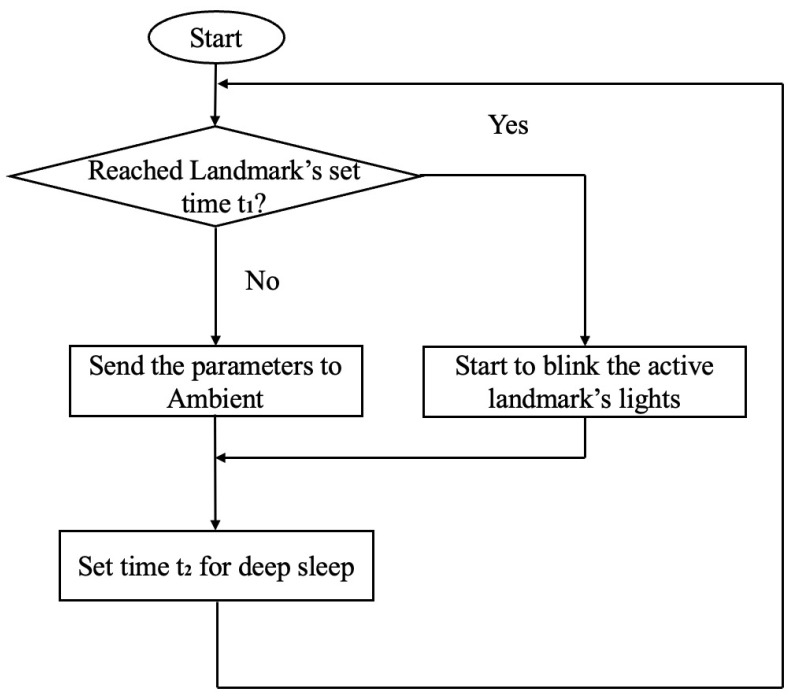
Operational flowchart of active landmark side.

**Figure 9 sensors-24-06152-f009:**
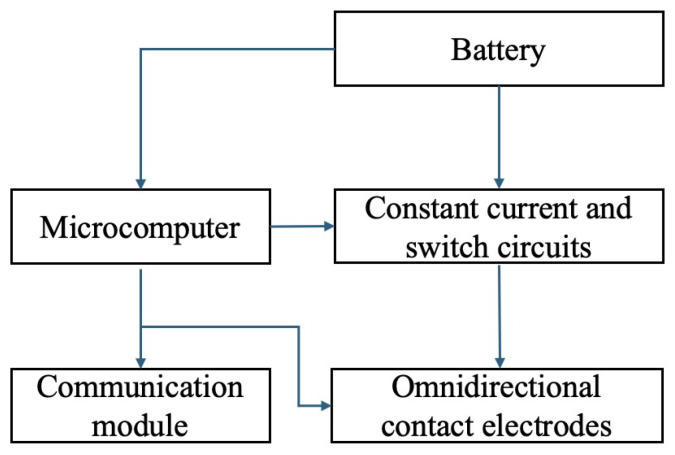
Constitution of the power supply side on mobile robots.

**Figure 10 sensors-24-06152-f010:**
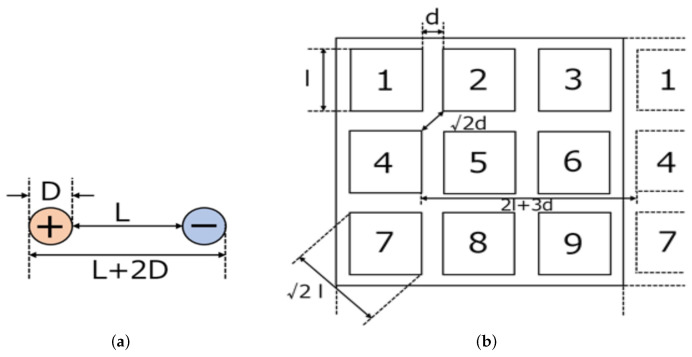
Design of the power supply and receiving sides. (**a**) Circular electrodes for power supply side. (**b**) Electrode pattern on the receiving side.

**Figure 11 sensors-24-06152-f011:**
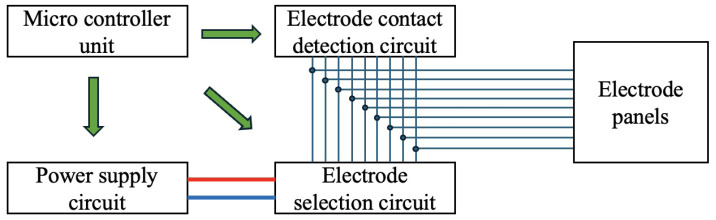
System configuration of the omnidirectional contact electrode.

**Figure 12 sensors-24-06152-f012:**
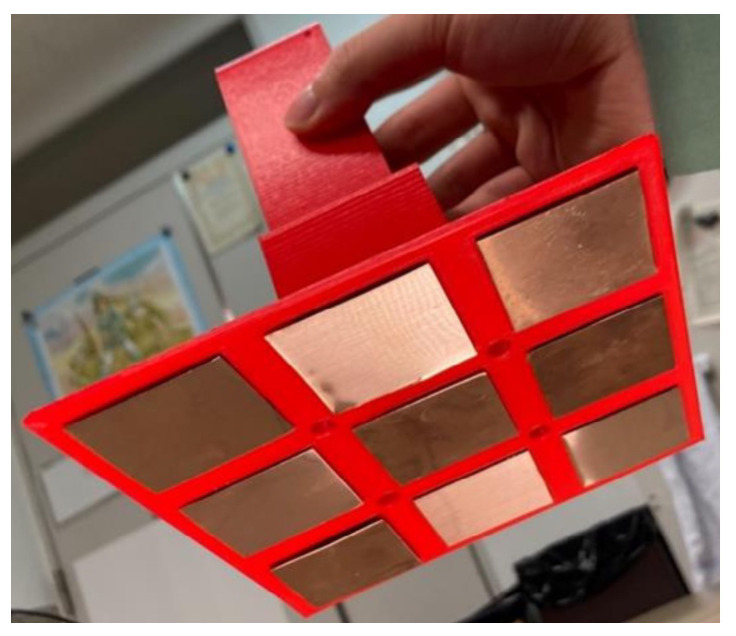
Omnidirectional contact electrodes.

**Figure 13 sensors-24-06152-f013:**
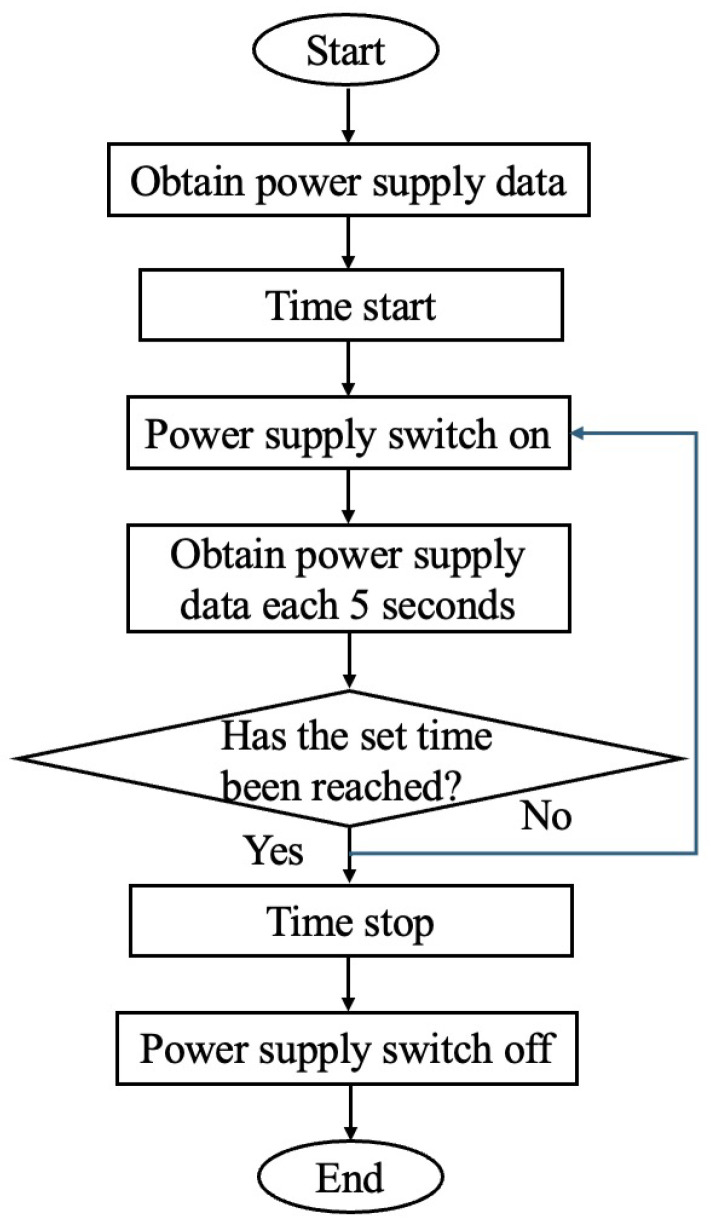
Operational flowchart of the power supply side.

**Figure 14 sensors-24-06152-f014:**
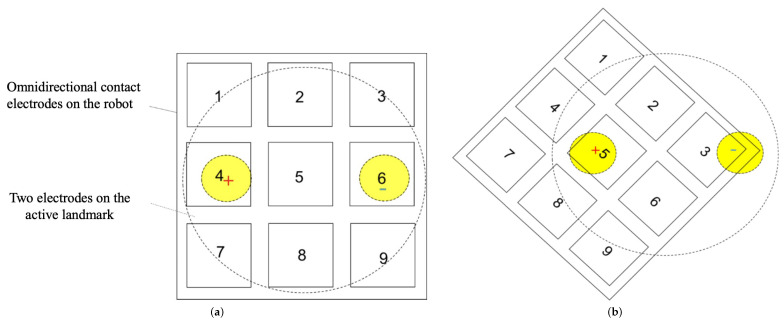
Contact cases between omnidirectional contact electrodes and active landmarks. (**a**) Case 1. (**b**) Case 2.

**Figure 15 sensors-24-06152-f015:**
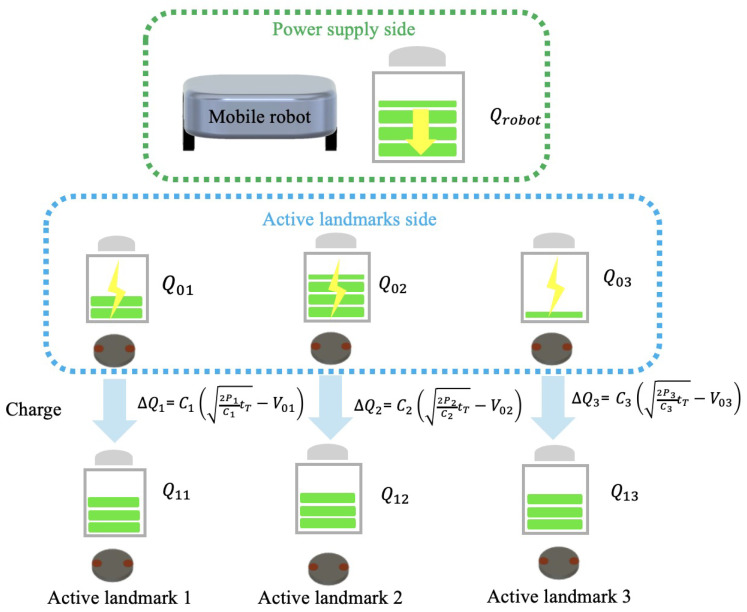
Illustration of the power distribution calculation module.

**Figure 16 sensors-24-06152-f016:**
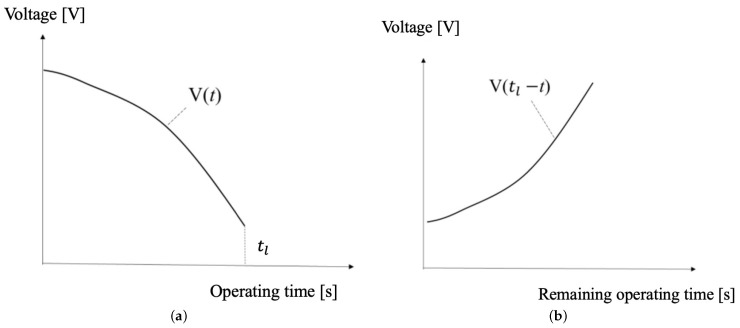
Operating voltage changes over time. (**a**) Relationship between landmarks’ battery voltage and operating time. (**b**) Relationship between landmarks’ battery voltage and remaining operating time.

**Figure 17 sensors-24-06152-f017:**
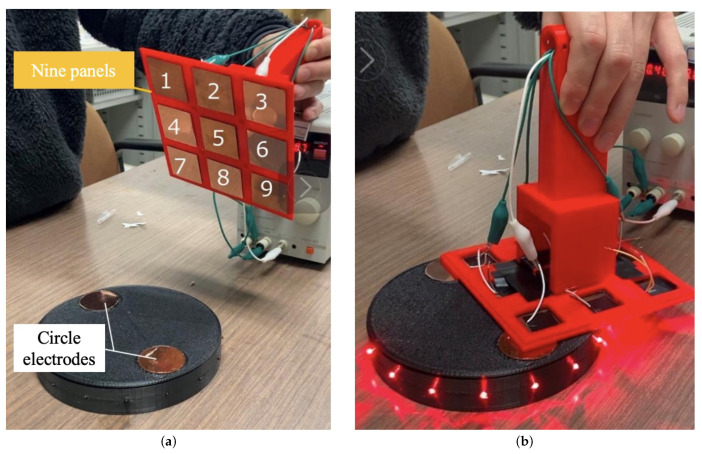
Contact between landmark side and omnidirectional electrodes on robot side. (**a**) Nine electrodes panel on robots and two circle electrodes on landmarks. (**b**) Contact cases between the landmark and robot.

**Figure 18 sensors-24-06152-f018:**
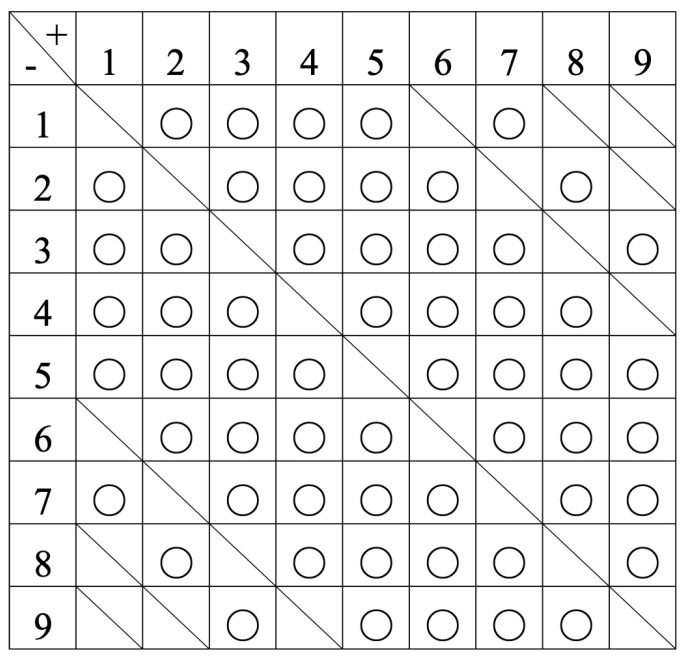
Patterns of connection status between omnidirectional electrodes and landmarks.

**Figure 19 sensors-24-06152-f019:**
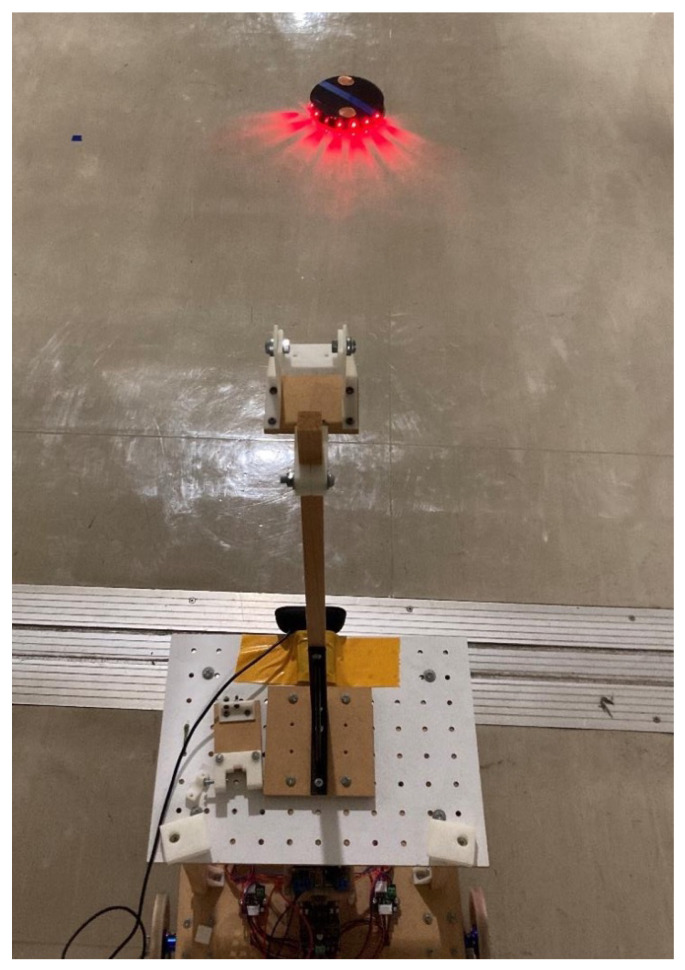
Detection of landmarks by the camera installed on the robot.

**Figure 20 sensors-24-06152-f020:**
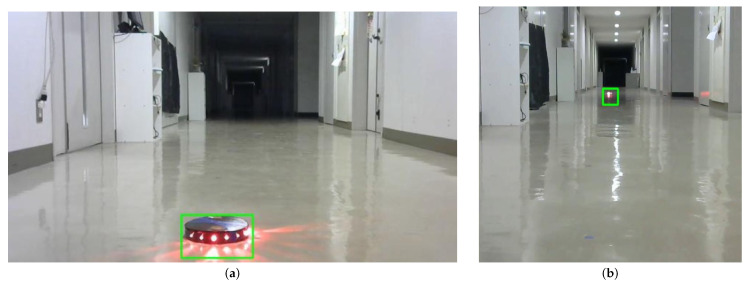
Landmark detection results at different distances. (**a**) Landmark detection at a distance of 1 m. (**b**) Landmark detection at a distance of 15 m.

**Figure 21 sensors-24-06152-f021:**
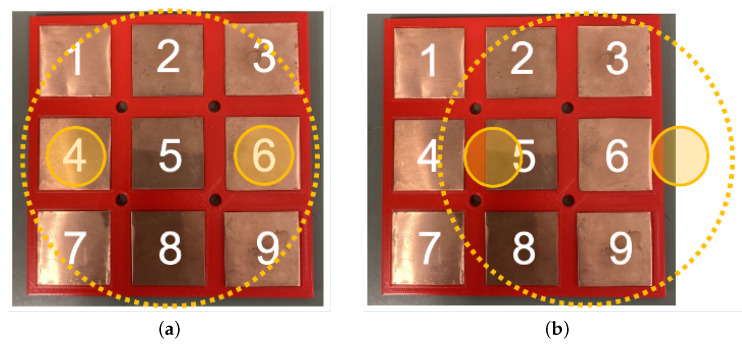
Error range of power-supplying and power-receiving sides for electrodes contacting successfully. (**a**) Power-supplying and power-receiving sides are perfectly aligned. (**b**) Largest error range for successful charging.

**Figure 22 sensors-24-06152-f022:**
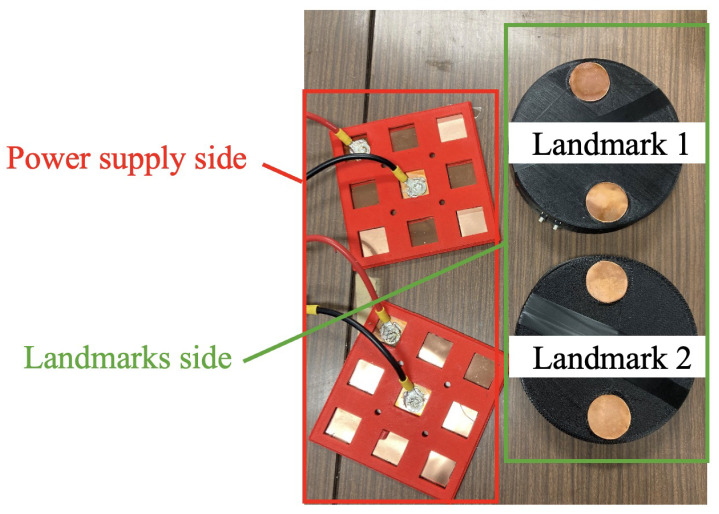
Power supply and landmark sides for controlled experiments.

**Figure 23 sensors-24-06152-f023:**
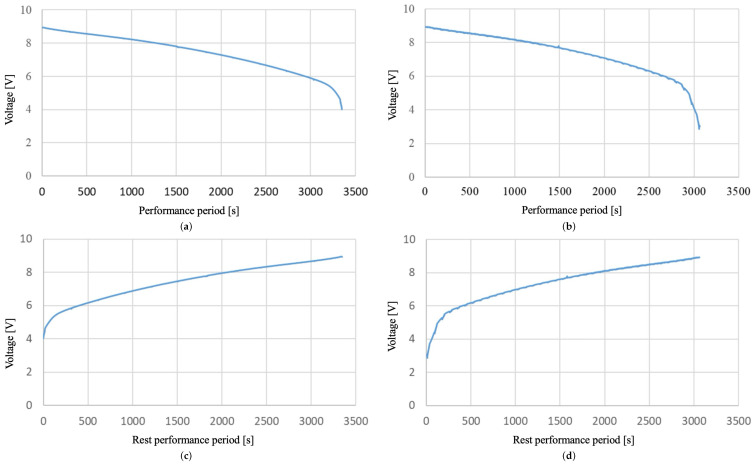
Control experiment for the proposed system persistence. (**a**) The relationship between elapsed time and battery voltage during the operation of landmark 1. (**b**) The relationship between elapsed time and battery voltage during the operation of landmark 2. (**c**) Remaining operating time and voltage of landmark 1. (**d**) Remaining operating time and voltage of landmark 2.

**Figure 24 sensors-24-06152-f024:**
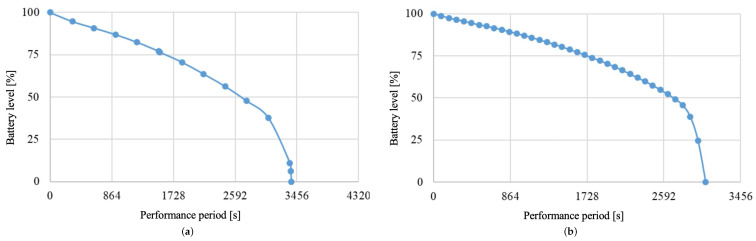
Battery information sent from landmarks to Ambient server. (**a**) Battery information sent from Landmark 1 to Ambient server. (**b**) Battery information sent from Landmark 2 to Ambient server.

**Figure 25 sensors-24-06152-f025:**
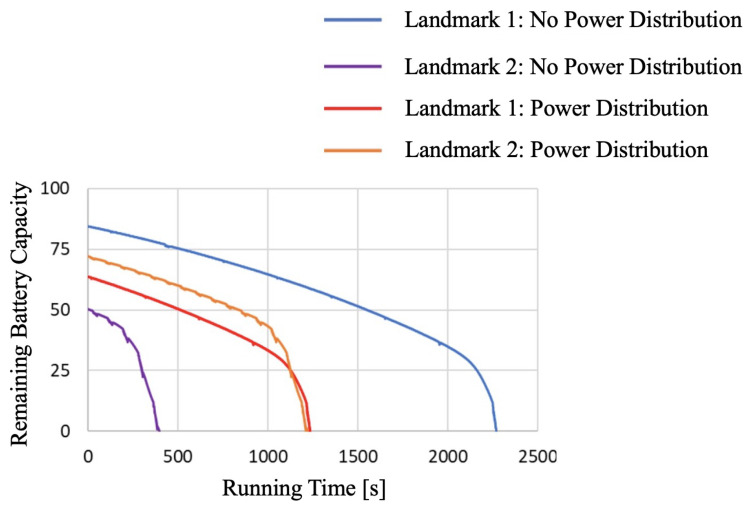
Elapsed time and battery voltage with and without power distribution adjustment.

**Figure 26 sensors-24-06152-f026:**
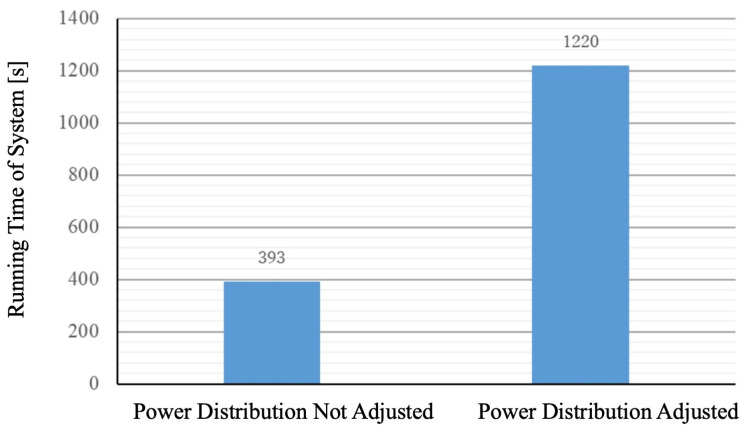
Comparison of system operating times with and without power distribution adjustment.

**Table 1 sensors-24-06152-t001:** Robot guidance movements using landmarks.

	The LED Lights Blink for Robot Navigation		
	**Navigation Frequency [min]**	**Blinks Period [ms]**	**Blinking Time [ms]**
Landmark 1	30	300	20
Landmark 2	1	400	20

**Table 2 sensors-24-06152-t002:** The battery level of the landmark before and after charging.

	Operating Time Attempting to Fully Charge [s]			Operating Time with Power Distribution [s]	
	**Landmark 1**	**Landmark 2**		**Landmark 1**	**Landmark 2**
Battery level before charging [%]	47.6	49.3		47.6	49.3
Battery level after charging [%]	84.4	51.9		59.2	65.7
Performance period theoretically [s]	2276	396		1261	1261

## Data Availability

The original contributions presented in the study are included in the article, further inquiries can be directed to the corresponding author.

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
