# Peer review of "An Adaptive and Automatic Power Supply Distribution System with Active Landmarks for Autonomous Mobile Robots"

_sensors, 2024, doi:10.3390/s24186152_

Round 1
Reviewer 1 Report
Comments and Suggestions for Authors
This paper focuses on miniaturizing active landmarks and optimizing power distribution among landmarks, which lowers the deployment cost of the landmarks and improves the system’s persistence. The active landmark system in this paper is divided into two parts: landmarks and robots. On the landmark side, the charging unit is simplified to only have positive and negative interfaces, which reducing the external volume of the landmark. On the robot side, charging unit is composed of 3 * 3 omnidirectional contact electrodes, which reduces the positioning accuracy requirement during charging. The adaptive and automatic power supply distribution system successfully minimizes installation constraints and costs while improving charging efficiency and operational duration. The technique presented in this paper is sound
However, the reviewer’s main concerns are given as follows.
1. The charging contact range between robots and landmarks is unclear. Although this paper indicates the size of the power supply and receiving sides, the actual usable range has not been calculated in detail. This range has an impact on the control positioning accuracy requirements of robots;
2. There are many unclear expressions in the article. For example, the expression of formula (4) is incorrect, Q should be the amount of charge in the landmark after charging. Is the of formulas 6 and 7 the same? If different, it is recommended to replace the identifier. And formula 8,10,11,12,15,16 all have such issues;
3. There is an incorrect statement between line 353 and line 356, Sec 4.1 and Sec 4.2 should be Sec 3.1 and Sec 3.2;
4. The image position should be slightly to the right, aligned with the text;
5. The design boundary defined in formulas 1, 2, and 3 is not perfect. D and L have only given a lower bound but not an upper bound, so D and L can be infinitely large;
6. Figure 24 and Figure 25 are repeated.
7. Figure 16’s X-axis and Y-axis are not clear.

The English expression is relatively clear, but there are some places that are a bit verbose
Reviewer 2 Report
Comments and Suggestions for Authors
Presented paper demonstrates adaptive and automatic power supply distribution system based on the active landmarks for autonomous mobile robots. Authors present interesting approach that can be adapted for practical use. Experiment is well designed and results described in details. There are only several comments according the paper:
1. Introduction is week and should be expanded with respect to the recent studies to highlight novelty of the presented research;
2. Table 2, operating time attempting to fully charge, landmark 2 – battery level before and after charging is the same 49,3% - there is no charging happens or this is misprint?
3. What material is used in contact electrodes? What contact resistance in landmark/robot electrode connection?
4. Please clarify in the text why omnidirectional contact electrodes were from the robot side.
Round 2
Reviewer 2 Report
Comments and Suggestions for Authors
Paper can be accepted for publication in the present form